# The highly dynamic nature of bacterial heteroresistance impairs its clinical detection

Cátia Pereira [1], Jimmy Larsson[2], Karin Hjort [1], Johan Elf [2] & Dan I. Andersson [1✉]

Many bacterial species and antibiotic classes exhibit heteroresistance, a phenomenon in which a susceptible bacterial isolate harbors a resistant subpopulation that can grow in the presence of an antibiotic and cause treatment failure. The resistant phenotype is often unstable and without antibiotic selection it reverts back to susceptibility. Here we studied the dynamics by which these resistant subpopulations are enriched in the presence of antibiotic and recede back to their baseline frequency in the absence of selection. An increasing understanding of this instability will allow more effective diagnostics and treatment of infections caused by heteroresistant bacteria. We show for clinical isolates of *Escherichia coli* and *Salmonella enterica* that different antibiotics at levels below the MIC of the susceptible main population can cause rapid enrichment of resistant subpopulations with increased copy number of genes that cause resistance. Modelling and growth rate measurements of bacteria with increased gene copy number in cultures and by microscopy of single-cells in a microfluidic chip show that the fitness cost of gene amplifications and their intrinsic instability drives their rapid loss in the absence of selection. Using a common antibiotic susceptibility test, we demonstrate that this test strongly underestimates the occurrence of heteroresistance in clinical isolates.

[1] Department of Medical Biochemistry and Microbiology, Uppsala University, Uppsala, Sweden. [2] Department of Cell and Molecular Biology, Uppsala University, Uppsala, Sweden. ✉email: dan.andersson@imbim.uu.se

Antibiotic resistance is a major public health concern[1]. Resistance is typically considered a stable trait emerging from either the acquisition of resistance genes by horizontal gene transfer or by mutations in native genes, but it also can be unstable and occur only in small fractions of a population, generating phenotypic heterogeneity within the bacterial population[2]. An example of this phenomenon is heteroresistance. First described in the 1940s, heteroresistance is a phenotype in which a bacterial isolate contains subpopulations of cells that exhibit increased antibiotic resistance in comparison to the main population[3–5]. Several experimental animal and clinical studies suggest that the enrichment of low frequencies of resistant bacteria can occur in infected hosts and lead to treatment failure[6–9].

Heteroresistance is highly prevalent for several different bacterial species and antibiotics[4,8] and often unstable. In some cases, the resistant subpopulations can revert to susceptibility after a limited number (<50) of generations of growth in the absence of antibiotic pressure[3,8,10,11]. Two different mechanisms explain this instability. The first mechanism involves the acquisition of resistance mutations that are genetically stable but confer a high fitness cost and the absence of antibiotics drives the selection of compensatory mutations that concomitantly reduce the cost and loss of resistance[8,12,13]. The second mechanism occurs when the subpopulation has an increased copy number of resistance-conferring genes due to tandem gene amplifications that are intrinsically unstable and costly, and consequently are lost in the absence of antibiotic selection[3,8,11,13]. A recent study demonstrated that heteroresistance caused by amplification of resistance genes is the most common type of heteroresistance in gram-negative bacteria[13]; it was found that from 766 bacterium–drug combinations tested, 27.4% exhibited heteroresistance and from these, 88% exhibited unstable heteroresistance associated with tandem gene amplification of known resistance genes[8].

The heteroresistance phenotype results from spontaneous amplification of a weakly expressed resistance gene, such as an antibiotic-modifying enzyme, whose increased dosage and over-production can generate a transient and resistant subpopulation[14,15]. Gene duplications are known to form through unequal crossing over, typically between directly repeated sequences such as insertion sequence elements, transposon genes, and ribosomal RNA operons[14,16]. The duplicated region then provides a perfect tandem repeat sequence that either can be deleted or further amplified[14,15]. In the presence of selective pressure for a gene(s) in the amplified region, cells with amplifications are favored. In the absence of selection for the amplified state, growth of revertants with reduced gene copy number is favored and the frequency of cells in the population with an amplification will reach a steady-state determined by the ratio of formation and loss and the fitness cost of the amplified unit[17,18].

Even though the basic properties of tandem amplifications are known, how selective pressures affect the formation, maintenance, and loss of heteroresistance remains poorly understood[14]. Of relevance in this context is recent work demonstrating the importance of low antibiotic concentrations for the selection and enrichment of resistant mutants. Thus, selection of either pre-existing or de novo formed resistant bacteria can occur at antibiotic concentrations that are several hundred-fold below the minimal inhibitory concentration (MIC) of a susceptible bacterium[19–21]. The minimal selective concentration (MSC) defines the lowest concentration of antibiotic at which a specific resistant bacterium will outcompete its susceptible counterpart and become enriched[19,22,23]. Since a heteroresistant population contains a mix of susceptible and resistant bacteria, determining the MSC for the resistant subpopulation of a heteroresistant isolate is a key parameter for understanding under which conditions the resistant subpopulations are enriched and lost in response to different antibiotic concentrations.

Here we examined the dynamics of gene copy number variation in heteroresistant bacteria from four clinical isolates of *E. coli* and *S.* Typhimurium against four different clinically used antibiotics (tobramycin, tetracycline, cefotaxime, and piperacillin-tazobactam) in order to determine: (i) if sub-MICs of antibiotics enrich for resistant subpopulations of bacteria with increased copy number, (ii) the fitness cost and (iii) the genetic basis of the resulting amplifications, (iv) the rates at which resistance and copy number is reduced in the absence of selective pressure, and (v) how this dynamic affects the detection of heteroresistance in clinical isolates. We show that sub-MIC levels of antibiotics can cause rapid enrichment of resistant subpopulations with increased copy number and that the fitness cost of gene amplification can be substantial and influence heteroresistance dynamics. This study demonstrates how the highly dynamic nature of heteroresistance impairs its detection and emphasizes the need for improved methods to rapidly detect and treat infections caused by heteroresistant bacteria.

## Results

**Sub-lethal antibiotic concentrations enrich for the resistant subpopulation in a concentration- and time-dependent manner.** To investigate the enrichment of subpopulations with increased antibiotic resistance in a susceptible main population, we evolved four clinical isolates of *E. coli* (strains DA33135 and DA61218) and *S.* Typhimurium (strains DA34827 and DA34833) by serial passage against four different clinically relevant antibiotics at concentrations equal or below the MIC of the parental strain. These strains, except DA61218, were previously shown to be heteroresistant to the tested antibiotics due to gene amplification of known resistance genes[8]. All strains were passaged for 160 generations in medium supplemented with different sub-lethal concentrations of each antibiotic (1/16x, 1/4x, 1/2x, and 1x the MIC of the parental strain). The enrichment of resistance was evaluated by measuring the average number of copies of the resistance gene within the population (included in the amplification) after 10, 20, 40, 80, and 160 generations of growth at 1x MIC and after 80 and 160 generations for the lower antibiotic concentrations (1/16x, 1/4x, 1/2x MIC). For all four strains, the resistance gene copy number increased with both exposure time and antibiotic concentration for concentrations as low as 16-fold below the MIC of the parental strain (Fig. 1). For all four antibiotics, the rate of gain of amplifications was fastest for the highest concentration, for which the accumulation of amplifications reached a plateau at 80/160 generations of selection. The maximum average number of copies acquired was dependent on the strain and antibiotic, ranging from 5 (piperacillin-tazobactam) to 13 (cefepime).

We showed by two independent tests that the increase in copy number was associated with increased resistance. First, for each antibiotic we chose one lineage evolved at the MIC of the parental strain and determined the fraction of resistant mutants by plating bacteria on 1x, 2x, 4x, 8x, and 16x, the MIC of the parental strain. Previous control experiments have shown that the amplifications are pre-existing and do not appear on the agar plates[8]. This population analysis profile (PAP) test showed that the enrichment of subpopulations with increased copy number was associated with a 16-fold increase in resistance as compared to the parental strain (Fig. 2, Table 1 and Supplementary Fig. 1). Heteroresistance is defined as the presence of a resistant subpopulation of cells at a frequency of $1 \times 10^{-7}$ or higher that grows at concentrations of antibiotic at least 8-fold higher than the concentration that does not affect the growth of the dominant

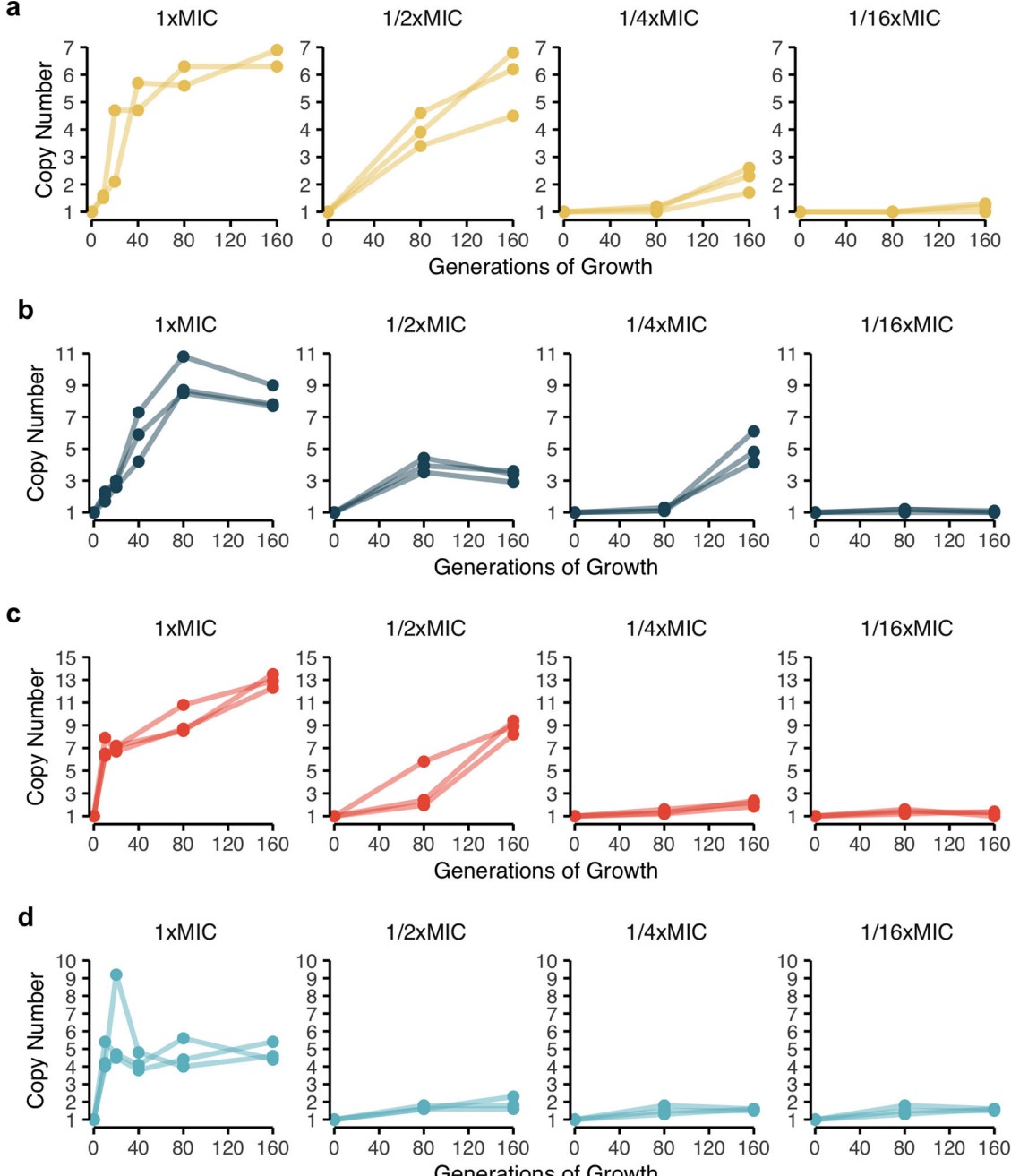

**Fig. 1 Gene copy number as a function of time and antibiotic concentration. a** *E. coli* DA33135 evolved in the presence of tobramycin, **b** *S.* Typhimurium DA34827 evolved in the presence of tetracycline, **c** *S.* Typhimurium DA34833 evolved in the presence of cefepime, and **d** *E. coli* DA61218 evolved in the presence of piperacillin-tazobactam. The antibiotic concentrations used for selection refer to the MIC of the parental strain. The gene copy number was determined after 10, 20, 40, 80, and 160 generations for the evolution in the presence of antibiotic at 1x MIC of the parental strain and after 80 and 160 generations for the remaining antibiotic concentrations in study. Two independent clones were analyzed for the evolution of *E. coli* DA33135 in the presence of tobramycin at 1x MIC of the parental strain; three independent clones were analyzed for all other experiments. The copy number was determined from the cell population. The copy number determination was based on 1 biological and 1 technical replicates ($n = 1$).

susceptible population[4,8]. Based on this definition, enrichment of resistant subpopulations in the heteroresistant isolates occurred for all antibiotics in the presence of antibiotic concentrations at 1x MIC of the parental strain after only 10 generations of growth, except for tetracycline, for which enrichment only occurred after 80 generations. After 160 generations, the fractions of mutants resistant to 8x and 16x MIC of the parental strain were equal to or higher than $10^{-3}$ and $10^{-4}$, respectively, for all antibiotics. Comparisons of the enrichment of resistant mutants over time

with the copy number increase for the same selective concentration showed a concomitant increase in gene copy number and resistance (Supplementary Fig. 2). As an additional test, we determined the MIC of the population for each strain and lineage evolved for 80 generations at 1/4 and 1x MIC of the parental strain (Supplementary Fig. 3). By comparing the population MIC with the MIC of the wild-type strain, we observed that for selection at 1/4xMIC, the population MIC increased 2- to 4-fold, whereas for selection at 1x MIC the increase was 8- to >2000-fold.

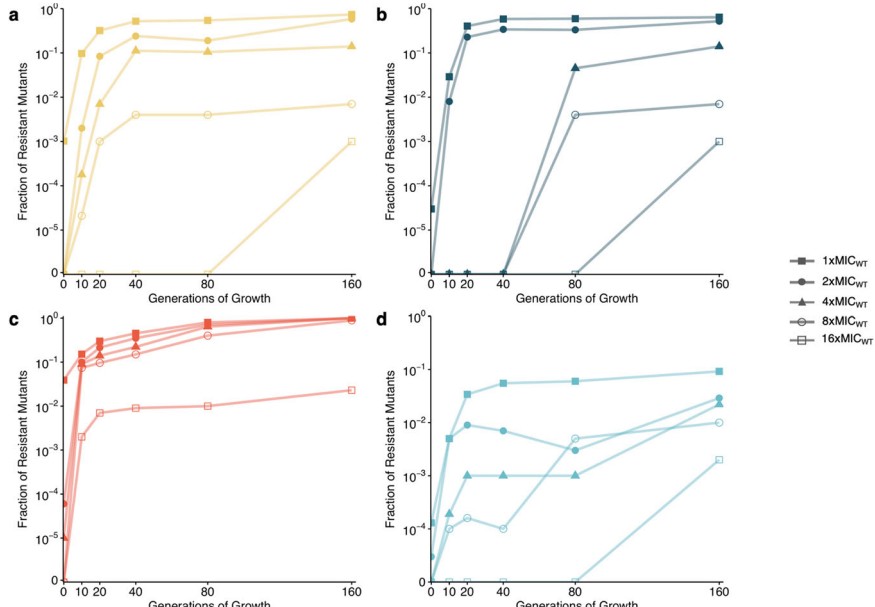

**Fig. 2 Fraction of resistant mutants of the subpopulations selected at 1x the MIC of the parental susceptible strain.** Subpopulations evolved from **a** *E. coli* DA33135 resistant to tobramycin, **b** *S.* Typhimurium DA34827 resistant to tetracycline, **c** *S.* Typhimurium DA34833 resistant to cefepime, and **d** *E. coli* DA61218 resistant to piperacillin-tazobactam. One serially passaged lineage per antibiotic was plated on Mueller-Hinton agar containing antibiotic at concentrations of 1x $MIC_{WT}$ (full squares), 2x $MIC_{WT}$ (full circles), 4x$MIC_{WT}$ (full triangles), 8x $MIC_{WT}$ (circles), and 16x $MIC_{WT}$ (squares) and the fractions of resistant mutants were calculated. Determination of fraction of resistant mutants was based on 1 biological and 1 technical replicates (*n* = 1).

**Table 1 Analysis of independent clones evolved for 80 generations at 1x MIC.**

| Antibiotic (starting strain) | Isolate (evolved strain) | MIC increase (x$MIC_{WT}$) | Relative growth rate | | Copy number | Amplification size (kbp) | Total DNA (kbp) | Fitness cost (%)/kbp of DNA | |
| --- | --- | --- | --- | --- | --- | --- | --- | --- | --- |
| | | | Batch culture | Single-cell | | | | Batch culture | Single-cell |
| Tobramycin (DA33135) | 1 (DA69068) | 16 | 0.96 ± 0.02 | 0.941 | 6.1 | 27.7 | 168.97 | 0.0237 | 0.0349 |
| | 2 (DA69069) | 16 | 0.93 ± 0.02 | 0.955 | 6.4 | 27.7 | 177.28 | 0.0395 | 0.0254 |
| | 3 (DA69070) | 12 | 1.00 ± 0.01 | 0.997 | 3.0 | 27.7 | 83.1 | 0.0001[a] | 0.0036[a] |
| Tetracycline (DA34827) | 1 (DA69071) | 8 | 0.88 ± 0.01 | ND | 10.9 | 9.7 | 105.73 | 0.1135 | ND |
| | 2 (DA69072) | 6 | 1.01 ± 0.01 | 0.992 | 7.0 | 9.7 | 67.9 | 0.0001[a] | 0.0118[a] |
| | 3 (DA69073) | 8 | 0.96 ± 0.01 | ND | 8.9 | 9.7 | 86.33 | 0.0463 | ND |
| Cefepime (DA34833) | 1 (DA69074) | 4 | 0.98 ± 0.01 | 0.975 | 8.2 | 10.5 | 86.1 | 0.0232 | 0.0290 |
| | 2 (DA69075) | 4 | 0.95 ± 0.01 | ND | 11.5 | 10.5 | 120.75 | 0.0414 | ND |
| | 3 (DA69076) | 6 | 0.91 ± 0.01 | ND | 14.7 | 10.5 | 154.35 | 0.0583 | ND |
| Piperacillin-Tazobactam (DA61218) | 1 (DA69077) | 32 | 0.96 ± 0.01 | ND | 9.9 | 3.5 | 34.65 | 0.1154 | ND |
| | 2 (DA69078) | 8 | 0.99 ± 0.03 | 1.002 | 5.3 | 3.5 | 18.55 | 0.0539 | −0.0027 |
| | 3 (DA69079) | 6 | 0.99 ± 0.01 | ND | 5.7 | 3.5 | 19.95 | 0.0501 | ND |

MIC was determined based on 3 biological and 1 technical replicates. Relative growth rates were determined based on 3 technical and 1 biological (batch culture) and 1 biological and 1 technical (single-cell) replicates.
*ND* not determined.
[a]These strains were not considered in the calculations of average cost per kbp of DNA since they had additional single mutations of unknown effects in their genome.

All strains reached MICs above the Clinical Breakpoints (Supplementary Table 1)[24].

Since sub-lethal antibiotic concentrations enriched for subpopulations of cells with gene amplifications and decreased susceptibility, we determined the lowest antibiotic concentration (MSC) that enriched for the resistant subpopulations. Since tandem gene amplifications are costly and intrinsically unstable, they are lost in the absence of selection[8,13,14,16]. In order for the amplifications to be maintained the fitness cost of the amplifications must be compensated by the growth advantage that the resistant subpopulations have in presence of antibiotic. By comparing the fitness (i.e., exponential growth rate) of the strain with a single gene copy and a clone with a gene amplification at

different antibiotic concentrations, it is possible to determine at which concentration the resistant bacteria have a competitive advantage over the susceptible population and are enriched, i.e., the MSC. For the clinical isolates of *E. coli* (DA33135) and *S.* Typhimurium (DA34827), we measured the relative exponential growth rate of the resistant strains (evolved for 20 and 80 generations) with an average copy number (CN) of 2.3 and 5.1 (DA33135) and 3.5 and 9.4 (DA34827), and the parental population (generation 0; CN = 1) at different antibiotic concentrations (Fig. 3a, b). For both antibiotics, the exponential growth rate of the resistant subpopulation is lower than that of the parental strain at lower antibiotic concentrations (Fig. 3a, b), but at higher concentrations, the resistant mutants grow faster

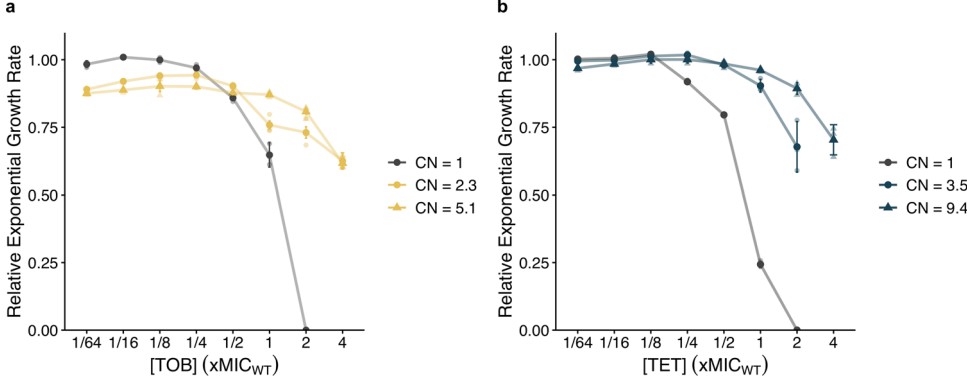

**Fig. 3 Relative exponential growth rates as a function of increasing concentrations.** Growth measured for independent clones selected in **a** tobramycin and **b** tetracycline for which copy numbers (CN) are 2.3 and 5.1 (in yellow circles and yellow triangles, respectively) and 3.5 and 9.4 (in blue circles and blue triangles), and the heteroresistant parental isolates with single copy (*E. coli* DA33135 heteroresistant to tobramycin and *S.* Typhimurium DA34827 heteroresistant to tetracycline, represented in gray). The growth rates are relative to the growth of the heteroresistant parental isolate in the absence of antibiotic. The standard deviations of three biological replicates are reported ($n = 3$).

than the parental strain. The crossover point of these two lines (gray and colored lines in Fig. 3) is the MSC[19], and as shown in Fig. 3a, b, it is 4- to 8-fold below the MIC. At several antibiotic concentrations, the difference in growth rate between the low copy number strain compared to the high copy number strain is statistically significant (Supplementary Table 2 and Supplementary Fig. 4). At concentrations below the MIC of the parental strain, the resistant isolate with lower copy number grew faster than the isolate with higher copy number. When selective pressure for the resistance phenotype increased (at concentrations above the MIC of the parental strain), the high copy number isolate grew significantly faster than the low copy number isolate.

**The fitness costs of tandem gene amplifications causing resistance.** A set of mutants with different gene copy numbers was used to determine the fitness effect of the extra DNA. Based on modeling it is expected that the fitness cost of an amplification will have a major impact on both the steady-state frequency of cells with amplifications and the level of amplification at a given selective pressure as well as the rate at which the amplifications are lost if antibiotic selection is reduced[17,18,25]. Ideally, pairwise competition between a single-copy and multi-copy strain for an extended time period would be the most sensitive method to determine fitness differences[19], but the high instability of the amplified state complicates such assays since the fitness cost changes over time. Instead, for three independent mutants per strain and per antibiotic, we determined growth rate in early exponential phase to reduce the number of cell divisions and thus minimize the effect of any potential copy number decrease on the measured fitness. Furthermore, from the same samples we determined the MIC, the copy number and identified the amplified region by whole-genome sequencing. The relative growth rates measured in batch culture ranged from $0.88 \pm 0.01$ (DA69071) to $1.01 \pm 0.01$ (DA69072) and the copy numbers ranged from 3.0 (DA69070) to 14.7 (DA69076). The size of the amplified units ranged from 3.5 (DA69077-79) to 27.7 (DA69068-70) kilo base pairs (kbp) and the total DNA within the amplified region (determined from the amplification size and copy number data) ranged from 18.6 (DA69078) to 177.3 (DA69069) kbp (Table 1). Based on these results, we calculated the average fitness cost per kbp of extra DNA to be 0.06% with a range from non-detectable to 0.12% depending on the specific amplification. The fitness costs in batch culture were confirmed by measuring single-cell growth (i.e., rate of cell size increase), which were determined by microscopy measurements of cells

captured in a microfluidics chip[26]. Measurements were recorded over a 30-min interval immediately after cell resuspension, to limit the time window (approximately one cell division) for any amplification to be lost by unequal crossing-over during replication and cell division (Supplementary Fig. 5). The estimated fitness cost per kbp of extra DNA ranged from non-detectable to 0.03% for the single-cell fitness measurements, suggesting that the loss rates were not sufficiently high to impact the fitness measurements in batch cultures (Table 1). Considering the small fitness effects measured (a few percent) and the differences in physiological conditions between batch culture and single-cell growth, a difference between the assay systems is not unexpected.

**Structure and location of tandem amplifications.** The WGS results showed that all amplifications included genes known to be involved in resistance toward the antibiotic used for selection (Table 2). The amplification breakpoints were located within directly repeated sequences, such as insertion sequences (IS6) for amplifications on plasmids and *qacE* for amplifications on the chromosome. For 10 out of the 12 mutants analyzed, no additional mutations were found, and the amplified units did not carry genes or loci known to be involved in copy-number regulation. The mutants DA69070 and DA69072, evolved in the presence of tobramycin and tetracycline, respectively, each had a single additional mutation in hypothetical genes. Since the potential effect of these mutations on fitness is unknown, they were not considered in the calculations of the average fitness cost per kbp of extra DNA presented above. Notably, no fitness cost was observed for these strains, possibly indicating that these mutations were acquired as compensatory mutations to reduce the fitness cost resulting from the accumulation of gene amplifications.

**Gene amplifications are unstable and lost in the absence of selection.** Gene duplications in bacteria are generally considered to arise from homologous or illegitimate recombination between repeated sequences in sister chromatids at replication. Duplication rates ($k_{dup}$) from a single copy have been measured as $10^{-6}$ to $10^{-2}$ per cell per generation in bacteria[18,27]. Further unequal crossover events within the duplicated region that either lead to further amplification or loss of higher-level amplifications can occur at rates ($k_{rec}$) between $10^{-3}$ and $10^{-1}$ per cell per generation[17], whereas the deletion of a duplicated copy and the subsequent return to the initial single-copy state can occur at rates ($k_{loss}$) between $2.5 \times 10^{-6}$ and $4 \times 10^{-2}$ per cell per

**Table 2 DNA amplifications of independent clones with different copy numbers.**

| Strain Antibiotic | Copy number | Size of the amplified unit (kbp) | Amplification breakpoint | Putative antibiotic resistance gene involved in hetero-resistance | Other antibiotic resistance genes included in the amplified region | Other mutations |
|---|---|---|---|---|---|---|
| *E. coli* DA69068 Tobramycin | 6.1 | 27.7 kbp | IS6-like element IS26 family transposase | *aac(3)-IId* | *sul1, sul2, dfrA17, aadA5, aph(6)-Id, aph(3'')-Ib, tet(A)* | - |
| *E. coli* DA69069 Tobramycin | 6.4 | 27.7 kbp | IS6-like element IS26 family transposase | *aac(3)-IId* | *sul1, sul2, dfrA17, aadA5, aph(6)-Id, aph(3'')-Ib, tet(A)* | - |
| *E. coli* DA69070 Tobramycin | 3.0 | 27.7 kbp | IS6-like element IS26 family transposase | *aac(3)-IId* | *sul1, sul2, dfrA17, aadA5, aph(6)-Id, aph(3'')-Ib, tet(A)* | Point mutation in hypothetical gene (Pro51Ala) |
| *S.* Typhimurium DA69071 Tetracycline | 10.9 | 9.7 kbp | *qacE* | *tet(A)* | *floR, aadA2* | - |
| *S.* Typhimurium DA69072 Tetracycline | 7.0 | 9.7 kbp | *qacE* | *tet(A)* | *floR, aadA2* | Point mutation in hypothetical gene (Phe330Leu) |
| *S.* Typhimurium DA69073 Tetracycline | 8.9 | 9.7 kbp | *qacE* | *tet(A)* | *floR, aadA2* | - |
| *S.* Typhimurium DA69074 Cefepime | 8.2 | 10.4 kbp | *qacE* | *blaCARB* | *tet(A), floR* | - |
| *S.* Typhimurium DA69075 Cefepime | 11.5 | 10.4 kbp | *qacE* | *blaCARB* | *tet(A), floR* | - |
| *S.* Typhimurium DA69076 Cefepime | 14.7 | 10.4 kbp | *qacE* | *blaCARB* | *tet(A), floR* | - |
| *E. coli* DA69077 Piperacillin-Tazobactam | 9.9 | 3.5 kbp | IS6-like element IS26 family transposase | *blaSHV* | - | - |
| *E. coli* DA69078 Piperacillin-Tazobactam | 5.3 | 3.5 kbp | IS6-like element IS26 family transposase | *blaSHV* | - | - |
| *E. coli* DA69079 Piperacillin-Tazobactam | 5.7 | 3.5 kbp | IS6-like element IS26 family transposase | *blaSHV* | - | - |

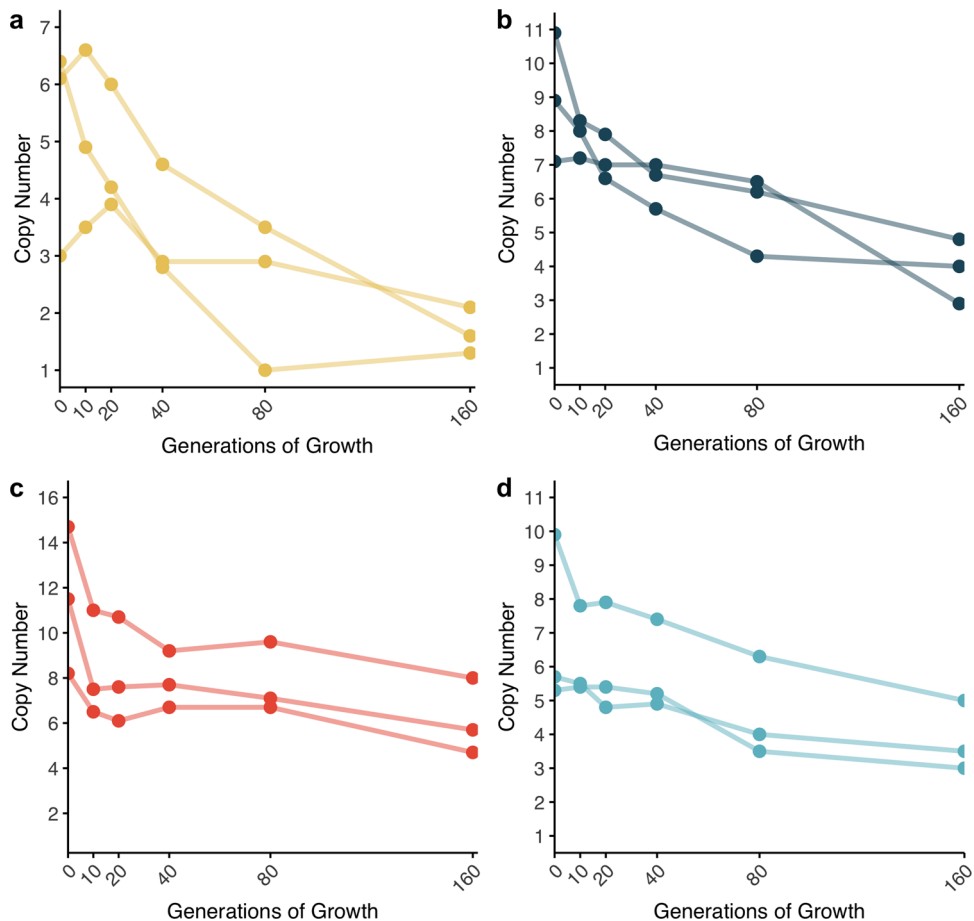

**Fig. 4 Gene copy number reduction as a function of time for mutants with amplifications grown in the absence of antibiotic. a** Tobramycin-resistant *E. coli* (DA69068-70), **b** tetracycline-resistant *S.* Typhimurium (DA69071-73), **c** cefepime-resistant *S.* Typhimurium (DA69074-76), and **d** piperacillin-tazobactam resistant *E. coli* (DA69077-79). For each antibiotic and strain, three independent clones were examined (see Table 1). Determination of copy numbers was based on 1 biological and 1 technical replicates ($n = 1$).

generation[15,17,18]. If the duplicated state is cost-free and in the absence of selection for the amplified state, the steady-state of duplications in the population will be determined by the ratio of the rate of duplication formation and loss, $\frac{[D]}{[H]} = \frac{k_{dup}}{k_{loss}}$. If there is a cost associated with the gene amplification, the ratio is set by $\frac{[D]}{[H]} = \frac{k_{dup}}{(k_{loss}+fitness\ cost)}$[17]. If an antibiotic selective pressure is applied the duplication can be enriched, and if antibiotic selection is removed the rate at which the duplications are lost will depend on the intrinsic rate of loss and the relative growth rate differences between the clones carrying different copy numbers.

To measure the loss dynamics of the amplified units, we used the bacteria with increased copy number (Table 1) as starting strains for a serial passage experiment performed in the absence of antibiotic selection and determined the relative number of copies of the resistance genes included in the amplified units after 10, 20, 40, 80, and 160 generations of growth. For all strains, an extensive loss of amplifications is seen after serial passaging with some variability depending on the starting copy number and specific resistance (Fig. 4). Based on the dynamics of gene copy number loss, we used the model for homologous recombination previously formulated by Pettersson et al.[18,25] to determine the recombination rate and theoretically inferred cost per additional copy (see "Methods"). The model was applied to the serial passaging experiment performed in the absence of antibiotic for the clinical strains in which no additional mutations were found (Supplementary Fig. 6) and the parameters $s$ and $k_{rec}$ were chosen

to best fit the rate of loss dynamics for these experiments. The range of parameters that best fitted the loss dynamics were an average cost per kbp of DNA of 0.06% and $k_{rec} \sim 0.003 \pm 0.0003$ (Supplementary Table 3). This cost estimated from an independent modeling approach is the same as the fitness cost directly measured from independent strains with different copy numbers (Supplementary Table 3). Based on the estimated $k_{rec}$, the calculated $k_{loss}$ was 0.012.

**The high instability of the resistant subpopulations impair detection of heteroresistance by disk diffusion antibiotic susceptibility test.** To examine how the high instability of the resistant subpopulation might influence its detection in clinical isolates, we performed a test mimicking clinical sampling and analysis of a blood infection. To this end, we used a set of 11 isolates of *E. coli* obtained previously from blood infections, and that showed unstable heteroresistance to gentamicin as demonstrated by the golden standard method for detection of heteroresistance, population analysis profiling (see "Methods"). In addition, two of the isolates studied in detail above, DA69074 (*S. enterica*, heteroresistant to cefepime) and DA69078 (*E. coli*, heteroresistant to piperacillin-tazobactam) were also analyzed.

The 11 isolates were initially tested by disk diffusion: heteroresistance was not detected by this method and all strains were classified as susceptible (Table 3). Subsequently, they were grown at 1x MIC of the respective antibiotic for 20 generations to enrich for the resistant subpopulations. This antibiotic exposure

**Table 3 Antibiotic susceptibility of heteroresistant isolates before and after growth at 1x the MIC of the parental strain and after growth in blood culture determined by disk diffusion (average diameter) and respective fraction of mutants resistant to 8x the MIC of the parental strain.**

| Strain | Antibiotic | Antibiotic Susceptibility Starting Strain (Average Diameter) | Antibiotic Susceptibility After Growth at 1xMIC$_{WT}$ (Average Diameter) | Antibiotic Susceptibility After Growth in Blood Culture (Average Diameter) | Fraction of Mutants Resistant to 8xMIC$_{WT}$ |
|---|---|---|---|---|---|
| DA34833 | Cefepime | S (28 mm) | R (23 mm) | S (27 mm) |  |
| DA61218 | Piperacillin-Tazobactam | S (24 mm) | R (12 mm) | - (18 mm) |  |
| DA63680 | Gentamicin | S (19 mm) | R (15 mm) | S (19 mm) |  |
| DA63702 | Gentamicin | S (22 mm) | R (16 mm) | S (18 mm) |  |
| DA63818 | Gentamicin | S (21 mm) | R (16 mm) | S (21 mm) |  |
| DA63834 | Gentamicin | S (21 mm) | R (15 mm) | S (17 mm) |  |
| DA63836 | Gentamicin | S (23 mm) | R (16 mm) | S (18 mm) |  |
| DA63870 | Gentamicin | S (22 mm) | R (16 mm) | S (18 mm) |  |
| DA63874 | Gentamicin | S (22 mm) | R (16 mm) | S (19 mm) |  |
| DA63882 | Gentamicin | S (20 mm) | R (16 mm) | S (18 mm) |  |
| DA63918 | Gentamicin | S (22 mm) | R (16 mm) | S (19 mm) |  |

Antibiotic susceptibility was determined based on 1 biological and 6 technical replicates and fraction of mutants resistant to 8x MIC of the parental isolate based on 1 biological and 1 technical replicates. S susceptible, R resistant.

would mimic the enrichment that would happen during antibiotic treatment in a patient. After this step, all isolates were classified as resistant based on the disk diffusion test. As a final step, the populations containing the enriched resistant subpopulations were inoculated into 10 mL of human blood to give 100 bacteria/mL. This mix was then transferred to blood culture flasks, grown for 12 h (corresponding to ~20 generations of growth) and then tested by disk diffusion. Strikingly, the resistant subpopulations

had been reduced and all 11 different isolates were now classified as susceptible. These findings were mirrored by the frequency of cells in the populations that showed a resistance of 8x the MIC of the parental strain (above the clinical breakpoints of the respective antibiotics) (Table 3). Thus, initially the fraction was low, after growth in presence of antibiotics it increased by 2 to 6 logs and after growth without antibiotic in blood culture flasks it was decreased 1 to 6 logs depending on the specific isolate.

## Discussion

Heteroresistance has a unique combination of attributes: a majority susceptible population unrestrained by resistance-associated fitness costs combined with a pre-existing, resistant subpopulation that can withstand an initial antibiotic exposure and replicate in the presence of the drug. In this study, the dynamics of enrichment and loss of subpopulations with decreased antibiotic susceptibility due to increased gene copy number was examined in four independent clinical isolates of two bacterial species. Serial passaging in the presence of sub-lethal antibiotic concentrations of four clinically relevant antibiotics showed that concentrations as low as 1/16 of the MIC of the parental strain could enrich for subpopulations with increased resistance gene copy number (Fig. 1). Furthermore, for tobramycin and tetracycline the minimal concentration (MSC) of antibiotic that can select for resistant subpopulations was well below the MIC of the parental strain (Fig. 3a, b).

These results are important since sub-lethal antibiotic concentrations are commonly found in human and animal tissues during therapeutic and/or growth promotion use of antibiotics as well as in many other types of environments[22,28,29]. Although antibiotics are usually administrated to result in plasma/tissue concentrations exceeding the MIC of the drug for the target bacteria, the antibiotic concentration might fall below the MIC[30–32]. This work demonstrates that antibiotic levels at MIC and sub-MIC can enrich the highly resistant subpopulations that potentially can cause treatment failure[6,7,9,33,34]. After relatively short exposure times, resistance levels considerably above the clinical breakpoint values were reached (Fig. 2 and Supplementary Fig. 3), suggesting enrichment of these subpopulations could be associated with high likelihood of therapeutic failure according to EUCAST guidelines[24]. Such enrichment could also occur in other environments, such as different aquatic environments and soils contaminated by anthropogenic activities[22,28,29].

Three different approaches, which showed good concordance, were used to determine the fitness cost of carrying and expressing extra DNA. Analysis of independent mutants carrying different copy numbers and measurements of batch culture growth rates and single-cell growth rates showed that the fitness cost per extra kbp of DNA ranged from non-detectable to 0.12%, with an average fitness cost of 0.06% for batch culture and 0.02% for single-cell measurements (Table 1). The observed variation in fitness cost per extra kbp of DNA for the different amplifications was expected since the deleterious effects of an increased gene dosage might vary due to the specific gene content of the amplification. Importantly, this cost has a major influence on the MSC and enrichment of resistant subpopulations; the higher fitness cost of the resistance (i.e., the gene amplification), the higher the antibiotic concentration (MSC) needed to confer enrichment and vice versa[19,35]. A previously formulated theoretical model[18,25] for homologous recombination was also used to calculate the cost of carrying extra DNA as well as the recombination rate given the experimental data on segregation of gene amplifications in the absence of selection. This deterministic model was used to identify the parameters that would best fit the empirical loss dynamics and a rather narrow range of parameter

values was found to agree with the data. The estimated value for the average cost was 0.06% per kbp of DNA, equal to the average cost determined from the batch culture growth rate data (Supplementary Table 3). Previous studies in *E. coli* and *Salmonella* strains show that for duplications in the range of around 10–1000 kbp, the cost per kbp ranges from $5 \times 10^{-5}$ to $1.5 \times 10^{-3}$, similar to the values determined here[15,18,25].

One of the main concerns associated with resistance caused by gene amplifications is their intrinsic instability, as the amplification and associated resistance can be rapidly lost after growth in absence of selection. During serial passaging of isolates with different initial gene copy numbers in absence of antibiotic, the reduction in copy number was higher than 50% after 160 generations for 6 out of the 10 mutants (Fig. 4). The average rate of loss per generation of growth calculated from our results was 0.31%. However, a previous study reported much higher variability, with the highest loss rate being 2.5% per cell per generation[18], thus suggesting that the loss rate is dependent on the specific amplification with regard to size as well as its gene content.

The impact of this instability due to the rapid loss of gene amplifications is evident from the analysis of bacteria that were enriched for the resistant subpopulation during growth in presence of antibiotic (as is likely to happen during treatment) and then cultured in blood culture flasks. Thus, isolates that were classified as resistant had after only 12 h of growth in blood cultures reverted to susceptibility (Table 3). This unstable nature of the resistant subpopulation was also confirmed by showing that the fraction of mutants resistant to 8x the MIC of parental strain (an antibiotic concentration which is equal/above the clinical breakpoint of the antibiotics studied[24]) was strongly increased after growth in the presence of antibiotic followed by a rapid decline after growth in blood cultures (Table 3). Overall, these results illustrate the highly dynamic nature of heteroresistance and the ease by which a resistant subpopulation can be enriched in presence of antibiotic and lost in its absence. The latter is of particular clinical importance since the enrichment of a resistant subpopulation of bacteria in blood of a patient, potentially causing treatment failure, could go undetected after a standard blood culture and antibiotic susceptibility testing, resulting in misdiagnosis of the bacteria as susceptible. Our work emphasizes the need for improved detection methods with continuous antibiotic selection during sampling and analysis as well as controlled clinical studies to better evaluate the impact of this dynamic on treatment failure. With regard to treatment, recent animal experiments show for carbapenem-resistant Enterobacteriaceae clinical isolates that combination therapy with multiple drugs could efficiently eradicate heteroresistant isolates[26].

## Methods

**Strains, growth conditions, and antibiotics**. The isolates used in this study are described in Supplementary Table 4. Throughout the study, Mueller-Hinton medium (Difco) was used as broth or in agar plates. Antibiotics (tobramycin, tetracycline, cefotaxime, piperacillin, and tazobactam) were purchased from Sigma-Aldrich. All incubations were performed at 37 °C, and liquid cultures were aerated by shaking (190 rpm).

**Experimental evolution of heteroresistant subpopulations**. To investigate if the enrichment of resistant subpopulations with increased gene copy number could be selected at sub-lethal antibiotic concentrations, clinical isolate cultures were serially passaged at concentrations of antibiotic correspondent to the MIC of the wild type, and at 2, 4, 16, 64, and 256-fold the MIC. A total of three independent lineages of each isolate were serially passaged at 1000-fold dilution in 2 mL batch culture every 24 h for 160 generations (10 generations of growth per serial passage) in Mueller-Hinton medium supplemented with antibiotics at the indicated concentrations (see Supplementary Fig. 7 for a schematic outline of the experiments and analysis). An aliquot of the culture was stored at −80 °C in 10% dimethylsulfoxide, after growth

for 10, 20, 40, 80, and 160 generations. In parallel, 340 μL samples from the same cultures were pelleted and stored at −20 °C for DNA extraction and subsequent gene copy number determination by digital droplet PCR (ddPCR).

**MIC measurements**. The MIC of the parental isolates was determined using Broth Microdilution in Mueller-Hinton broth. Bacterial strains were isolated on Mueller-Hinton agar plates and three colonies were resuspended in 0.9% NaCl to Mac-Farland 0.5 (determined using a Sensititre Nephelometer, Thermo Fisher Scientific) and diluted 1:10,000 in Mueller-Hinton broth (to ~$10^4$ cells/mL). Aliquots of 50 μL were mixed with 50 μL of medium containing serially-diluted antibiotics in 96-well round-bottomed plates. MICs were determined after incubation at 37 °C for 18 h. The MIC of the isolates with variable copy number (see "Characterization of isolates with variable initial copy number") was measured by Etest strips (Bio-Merieux) for tobramycin, tetracycline, and cefotaxime. Overnight cultures of the bacterial isolates were diluted 1:20 in PBS (8 g/L NaCl, 0.2 g/L KCl, 1.44 g/L Na$_2$HPO$_4$, and 0.24 g/L KH$_2$PO$_4$) and the cell suspensions were evenly spread onto agar plates using sterile cotton swabs. The Etest strips were applied and the plates were read after 24 h of incubation at 37 °C. For piperacillin-tazobactam, the MIC was determined using Broth Microdilution as previously described, using an initial culture inoculum of $10^6$ cells/mL.

**Characterization of isolates with variable initial copy number**. For each of the four strains, one isolate from each independent lineage serially passaged for 80 generations at the MIC of the parental strain was further analyzed (Supplementary Fig. 7). The bacterial strains saved at −80 °C were isolated on Mueller-Hinton agar plates with antibiotics (at the same concentration on which they were selected). After 12 h of growth at 37 °C, one colony was resuspended in 0.9% NaCl to 0.5 MacFarland. From this resuspension, two aliquots of 10 μL were used to start the batch culture and single-cell fitness cost measurements. Also, from the same cell suspension, two independent overnight cultures were inoculated, one in the absence of antibiotics to determine the stability of the resistant population and one in the presence of antibiotics for later DNA extraction. After MIC determination, the remaining fraction was pelleted and stored at −20 °C for later DNA extraction and determination of the gene copy number by ddPCR.

**Determination of gene copy number**. The DNA of the strains analyzed for gene copy number was isolated using bead-beating. The DNA extraction was performed from cultures with cell count of ~$3.4 \times 10^8$ cells. Cell disruption was performed by resuspension of the pelleted cell culture in 250 μL 1:1 water: Fast Lysis Buffer for Bacterial Lysis Solution (QIAGEN) with 0.75 g of 212–300 μm acid-washed glass beads (Sigma-Aldrich), followed by addition of 250 μL of Phenol: Chloroform: Isoamyl Alcohol (Sigma-Aldrich) and two cycles of bead-beating for 20 s at 6.5 m/s on a FastPrep-24™ Classic Instrument (MP Biomedicals). The gene copy number was determined via ddPCR using the QX200™ ddPCR system (Bio-Rad) according to the manufacturer's recommendations. The ddPCR reactions were prepared using the 2x EvaGreen Supermix (Bio-Rad), DNA template, and HindIII-HF (NEB) restriction enzyme (10 U/μg of DNA sample). Primers (100 nM) were added into each reaction. The list of primers used in this study can be found in Supplementary Table 5. An aliquot of 20 μL was taken from the 22 μL PCR reaction mix and loaded into a DG8 cartridge with the Droplet Generation Oil for EvaGreen (Bio-Rad). PCR was performed as follows: initial enzyme activation at 95 °C for 10 min, denaturation at 94 °C for 30 s, and amplification at 58 °C for 2 min over 50 cycles, followed by enzyme deactivation at 98 °C for 10 min. For all steps, a ramp rate of 2 °C/s was used. Following PCR amplification, samples were analyzed on the QX200 Droplet Reader using the QuantaSoft™ software (Bio-Rad). Samples with droplet counts below 10,000 droplets were discarded.

**Batch culture fitness cost measurements**. Growth rates of batch culture were analyzed in a Bioscreen C apparatus (Oy Growth Curves AB, Ltd) using the cell suspensions prepared as previously described (see "Characterization of isolates with variable initial copy number"). Bacterial cultures (1 μL) were used to inoculate 300 μL cultures that were grown for 24 h at 37 °C in the Bioscreen apparatus. Absorbance (A$_{600nm}$) was measured every 4 min, with cultures shaken between measurements. Absorbance values between 0.02 and 0.09 were used to calculate the exponential growth rate. The relative growth rates were determined by normalizing the growth rates to the growth of the parental strain, which was set to 1.

**Single-cell fitness cost measurements**. In order to measure the fitness cost of single-cell samples with increased copy number in a microfluidics set-up, a Ti2-E microscope (Nikon) equipped with a Plan Apo Lambda 100× objective (Nikon) was used. For imaging, a light-emitting diode light source (Nikon) together with an external phase module was used and all images were captured on a DMK 38UX304 camera (TheImagingSource). During experiments, the temperature was kept at 37 °C using a temperature-controlled enclosure (Okolab). The evolved mutants isolated on Mueller-Hinton agar plates (see "Characterization of isolates with variable initial copy number") were resuspended in filtered Mueller-Hinton medium supplemented with 0.017% (wt/vol) Pluronic F-108 (Sigma-Aldrich). Loading was done using media/cell containers connected to a pressure regulator (Elvesys). Control and sample cells were loaded into 1000-nm-wide traps in separate

channels in a PDMS microfluidic chip. Loading was done for the two cell conditions using port 2.1 and port 2.2 to keep the cells in separate compartments, as previously described[36,37]. After cell loading, the cell-containing tubes connected to the chip port 2.1 and port 2.2 were removed and exchanged for tubes with Mueller-Hinton medium supplemented with pluronic, supplied at a pressure of 200 mbar. Cell imaging started within 15 min after initiation of loading and up to 90 positions were imaged in phase contrast every minute. The code used for image analysis is described in Camsund et al. and Lawson et al.[37,38]. For estimation of single-cell growth rate, we used the formula $y = a*\exp(b*t)$, where $y$ is the cell area, $b$ is the growth rate, and $t$ is the detection time; $b$ was found by using the MATLAB function fit('exp1'). Cells were sorted into time bins based on the median of their detection frames.

**Population analysis profiling**. For one of the independent lineages serially passaged at the MIC of the parental strain for each strain in the study, a population analysis profile (PAP) test was performed. The same test was also performed for the parental strains (at generation 0). From the overnight cultures saved at −80 °C after 10, 20, 40, 80, and 160 generations of growth, $10^{-1}$ and $10^{-3}$ dilutions were prepared in Mueller-Hinton medium. The cell suspensions were then used to plate an appropriate amount of cells onto agar plates with antibiotic concentrations of 1, 2, 4, 8, and 16x MIC (of the parental strain). The plates were incubated at 37 °C for 24 h. The fraction of resistant mutants was calculated by dividing the population of bacteria growing in the presence of antibiotics by the total number of cells plated.

**Stability of the resistant population**. The stability of the resistant population was examined using the isolated clones from which increased copy number was determined. The overnight cultures saved at −80 °C were serially passaged at 1000-fold dilution (2 μL in 2 mL of Mueller-Hinton broth in the absence of antibiotics) every 24 h for 160 generations (10 generations of growth per serial passage). An aliquot of the culture was stored at −80 °C in 10% dimethylsulfoxide, after growth for 10, 20, 40, 80, and 160 generations. In parallel, 340 μL samples from the same cultures were pelleted and stored at −20 °C for future DNA extraxtion and determination of the gene copy number by ddPCR.

**Whole-genome sequencing (WGS)**. Samples with variable initial copy number were further investigated with Illumina WGS sequencing. Each isolate was used to start an overnight culture in 1 mL of Mueller-Hinton broth with antibiotic selection. From each culture, an aliquot of 100 μL was stored at −80 °C in 10% dimethylsulfoxide and the remaining culture was pelleted and stored at −20 °C for future DNA extraction and WGS. DNA was isolated using the MasterPure Comp kit (Epicenter), resuspended in dH$_2$O and concentrations were determined using the Nanodrop 1000 (Thermo Fisher Scientific) and Qubit 2.0 fluorometer (Invitrogen). Illumina sequencing was performed by Novogene (UK). Sequences were mapped to the reference genome using CLC Genomics Workbench software (Qiagen) and mutations were detected using the same software. Potential antibiotic resistance genes were determined with ResFinder using the default parameters (90% threshold and 60% minimum coverage)[39]. The reference genomes were submitted to the National Center for Biotechnology Information (NCBI) database and were automatically annotated using the Prokaryotic Genome Annotation Pipeline at the NCBI.

**Modeling of amplifications to calculate fitness cost and recombination rate**. In brief, the parameters affecting the frequency of neutral duplications in a population are dependent on the rate at which new duplications occur, $k_{dup}$, and the rate of homologous recombination, $k_{rec}$, that can lead to either an increase or decrease in copy number. Assuming that all copies are identical for the purpose of recombination and that the rate of recombination is constant per copy, the probability of recombination per cell division will increase with the number of copies ($m$) and be given by:

$$p_{rec}(m) = \frac{k_{rec}(m-1)}{1 + k_{rec}(m-1)} \tag{1}$$

This assumption considers that cells with higher initial copy number are more likely to undergo a recombination event. Moreover, once a recombination event occurs, the probability of going from an initial copy number ($n$) to any given final new copy number ($m$) in each recombination event is:

$$p(m|n) = \begin{cases} \frac{2m-n}{m^2} & \text{if } m \geq n \\ \frac{n}{m^2} & \text{if } n < m \end{cases} \tag{2}$$

This implies that a duplication is lost with rate ~$k_{rec}/4$, since $k_{loss} = p_{rec}(2)p(1|2)$. Furthermore, it is assumed that the change in relative growth rate for an individual that carries an extra copy of the gene(s) is given by $s$ and, last, that the fitness of an individual is negatively correlated to the number of extra copies carried. Thus, the fitness of a population with $m$ copies relative to the wild type with only one copy is:

$$w_m = (1 + s)^{m-1} \tag{3}$$

Considering that in a population there are several subpopulations with different numbers of amplifications ($m$), in fractions $f_m$ of the main population, the average instantaneous fitness of that population is given by the individual contribution of

the fitness of each fraction carrying different number of copies:

$$W = \sum_m w_m f_m \qquad (4)$$

Using this recombination model and given that each subpopulation with different copy number will compete according to their different relative fitness costs, the initial fraction of cells in a population carrying $m$ copies of an amplified region is given by:

$$A = \left(\frac{w_m}{W} - 1\right) f_m \qquad (5)$$

Moreover, given the probability of an initial fraction of the population with $n$ copies to change its copy number to $m$, the total rate of change in the fraction of cells that carries $m$ copies due to recombination events in all cells can be calculated by:

$$B = \sum_{m=2}^{\infty} p_{rec}(m) p(m|i) \frac{w_m}{W} f_m \qquad (6)$$

Finally, the reduction in the fraction of cells that have copy number $m$ is given by:

$$C = p_{rec}(m) \frac{w_m}{W} f_m \qquad (7)$$

Thus, in each generation, the fractions of the population with $m$ copies will change by:

$$\triangle f_m = A + B - C \qquad (8)$$

This is a fully deterministic model based on the following assumptions:

i. the population is very large and the changes in copy number occur fast since the efficiency for selection is increasingly relevant in larger populations;

ii. recombination can lead to an increase or decrease in copy number with equal probability and the rapid loss of amplifications is mainly driven by the fitness cost which limits the number of copies that can be found in a population (a high number of copies would result in an unreasonably high cost for the cell).

**Diagnosing heteroresistance using disk diffusion**. A total of 11 clinical isolates of *E. coli* isolated from positive blood cultures at Uppsala University Hospital during 2014–2015 that were confirmed to be heteroresistant (by PAP tests) were examined. They were subjected to susceptibility testing using the standard disk diffusion method, according to the EUCAST guidelines[24]. The gentamicin (Oxoid, 10 µg), cefepime (Oxoid, 30 µg) and piperacillin-tazobactam (Oxoid, 30–6 µg) disks were purchased from Thermo Scientific. A total of 4–6 disks were used per clinical isolate.

To determine the fraction of resistant mutants, appropriate dilutions of cell suspensions were spread on agar plates without antibiotics (to estimate total CFU) and at an antibiotic concentration of 8x MIC of the parental strain (to estimate the resistant CFU). The plates were incubated at 37 °C for 24 h. The fraction of resistant mutants was calculated by dividing the population of bacteria growing in the presence of antibiotics by the total number of cells plated.

To enrich for the resistant subpopulation clinical isolates were serially passaged at 1x the MIC of the wild type by transferring 2 µl of cells (~$10^6$ cells) into 2 mL of Mueller-Hinton medium supplemented with antibiotics at the indicated concentration and grown for 24 h. This was repeated once, resulting in a total of 20 generations of growth in the presence of antibiotic. The evolved cultures were then tested for antibiotic susceptibility using disk diffusion and fraction of mutants resistant to 8x the MIC of the parental strain was determined as described. These cultures were subsequently used for inoculation of the blood cultures. The blood culture bottles (BacT/Alert SA; bioMérieux) were filled with 10 mL of blood supplemented with ethylenediaminetetraacetic acid dipotassium salt dihydrate (EDTA) collected at Uppsala University Hospital to which 1 mmol/L of $MgSO_4$ and $CaCl_2$ was added. The blood culture bottles were inoculated with 1000 cells (mimicking a blood infection with 100 bacteria per mL of blood) and incubated at 37 °C for 12 h. The bacteria grown in blood cultures were then tested for antibiotic susceptibility by disk diffusion and fraction of mutants resistant to 8x the MIC of the susceptible strain as described.

**Statistics and reproducibility**. The details about the number of replicates and statistical analysis performed in this study are included in each figure legend. Comparisons between groups were determined using Student's $t$ test. Differences were considered significant at $p$-value <0.05 (*) and $p$-value <0.01 (**). Statistical analysis were made using R (v. 3.6.0).

**Reporting summary**. Further information on research design is available in the Nature Research Reporting Summary linked to this article.

## Data availability

Chromosomes and plasmids are deposited at NCBI under the following accession numbers: DA33135 (CP029576–CP029578), DA34827 (CP029593 and CP029594), DA34833 (CP029595 and CP029596), and DA61218 (CP061206 and CP061207). Data are available upon request.

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

## Acknowledgements
We thank Matthew Lukenge for providing PacBio sequencing data for strain DA61218 and Spartak Zikrin for data analysis of the single-cell fitness cost measurements.

## Author contributions
C.P. and D.A. designed the study. C.P. performed experiments and data analysis. J.L. performed single-cell fitness cost measurements. K.H. performed heteroresistance confirmation of the clinical isolates of *E. coli* isolated from positive blood cultures. C.P. and D.A. wrote the manuscript. All authors read and approved the manuscript. Supervision and funding acquisition was done by D.A. and J.E.

## Funding

## Competing interests
The authors declare no competing interests.
