## [Peer Review File · Communications Biology]

Reviewers' comments:

Reviewer #1 (Remarks to the Author):

The authors present a timely and important manuscript focused on the dynamics of antibiotic resistant populations of cells exhibiting resistance gene amplifications. This group has been instrumental in showing that gene amplification leads to heteroresistance but the dynamics of these amplifications have not previously been studied systematically. This manuscript is viewed as being an important and foundational building block for the field.

Major comments

1) Figure 3A – there is a trend where the growth rate is lower with higher gene copy number. Are these curves statistically different?

2) Is there a reason that the beta-lactams were not included in Figure 3?

3) Are the MICs for each strain and the relevant antibiotic listed anywhere? It would be interesting to know how they relate to the breakpoint for that given antibiotic. It seems that these are all strains with MICs below the breakpoint.

Minor comments

1) Is table 2 necessary? Every clone for each strain/antibiotic pair has the same amplification. The relevant information seems to be in Table 1 already.

2) The term “susceptible strain” is used often. This may be confusing or misleading to readers since the strains are known to contain resistant cells. Perhaps the term “parental strain” or “pre-treatment culture” would be better? It is important to choose a term that does not give the impression that there are no resistant cells present.

3) Figure 2 – the y-axes say “resistant mutants”. This is not a critical point but are the authors referring to resistant cells? If so, are all the resistant cells considered mutants? If variation in copy number among individual cells is the standard for certain strains, maybe the resistant cells with higher copy number should be viewed as some form of variant rather than a mutant? In these cases there are no point mutations or acquisition of genes, but just a change in copy number of existing genes.

Reviewer #2 (Remarks to the Author):

In this manuscript, the authors sought to examine the phenomenon of heteroresistance and its related dynamics. Understanding the heteroresistance dynamics, development and perpetuation of resistant subpopulations, will allow for better diagnostics and clinical treatment of patients harboring heteroresistant bacteria. The manuscript is addressing an important gap in the understanding of E.coli and S. enterica clinical disease and is well positioned to demonstrate dynamics under different drug pressures.

Overall this is a well executed study, addressing several aspects of heteroresistance dynamics in two different organisms and several different drugs illustrating possible farther reaching impacts. The manuscript could be strengthened by streamlining sample processing, with so many different sections it is confusing what isolate (or evolution of the isolate) the experiments are run on. The addition of a sample workflow might help clarify, as there are several sections where partial sample is saved, and some extracted for analysis at several of the same time points. The introduction is well laid out and clear with 5 main objectives, the results section has 6 components. The manuscript could be strengthened by having the results broken up in the same 5 objectives described in the last paragraph of the introduction. Lastly, in the discussion section could be strengthened by referring back to tables and figures that contain the results that are being discussed. Overall, this manuscript utilizes several different laboratory approaches, resulting in a well supported description of the instability of heteroresistance in these two organisms.

Minor

1. Table 1 – add units on total DNA column heading

2. Table 3 – add definitions of R/S/- in table legend
3. Figure 1 – add additional time points for the 1xMIC in figure legend
4. Figure 3 – need table legend that describes the shorthand (CN= Copy number) and add color depictions for greater impact of figure
5. Figure 4 – does not appear to be reference within manuscript text
6. Supplemental figures are not numbered in the order of appearance, sub fig 1 &2 are referenced in methods
7. Supplemental figure 4 – missing figure number and title
8. Supplemental figure 5 – add to figure legend what each color represents (yellow, blue, red, turquoise represent *E. coli* DA33135 resistant to tobramycin, *S. Typhimurium* DA34827 resistant to tetracycline, *S. Typhimurium* DA34833 resistant to cefepime and *E. coli* DA61218 resistant to piperacillin-tazobactam respectively
9. Result section - Within text can refer to Fig. 3 and remove AB designation (line 142 and 144)
10. Results section - Additionally add color into the description in the text to aid in reader's conclusion. Something like – " The crossover point of these two lines (gray versus colored lines in Fig. 3) is the MSC..." (line 143-144)

Reviewer #3 (Remarks to the Author):

It is well known that heteroresistance – a mixed population of susceptible and resistant cells within a bacterial isolate – can cause treatment failure as the resistant subpopulation can outcompete the susceptible subpopulation in the presence of antibiotics and is a major public health concern with some bacterial pathogens. This study characterizes the dynamic nature of these subpopulations within four heteroresistant clinical isolates and provides convincing evidence that: 1) not only can the resistant subpopulations be enriched in the presence of antibiotic, but they can also be reduced in the absence of antibiotic pressure, and 2) the enrichment can occur with antibiotic levels far below the MIC. The study focuses on a specific mechanism of heteroresistance, "an increased copy number of resistance-conferring tandem gene amplifications that are intrinsically unstable and costly". While my experience with heteroresistance is in *M. tuberculosis* which doesn't exhibit this mechanism, previous work by the authors has shown it to be the most common type of heteroresistance in gram-negative bacteria. The study tests multiple concentrations (down to 1/16x MIC) of antibiotic and time points (up to 160 generations) of bacterial growth, and get consistent results showing that even the lowest concentrations of antibiotic will increase the copy number of resistance genes within the population, and further show by two different methods that the increased copy number is associated with increased resistance. They then show that the copy number is reduced again using a serial passage experiment in the absence of antibiotic. Additionally, they characterized the structure and location of these tandem mutations and calculated the fitness cost of them, and also show that this instability in the population dynamics can confound the detection of resistance using the standard lab assay. While I am primarily a bioinformatician and can't speak to the specifics of the laboratory methods, they are very detailed and clearly worded which should make the results easy to reproduce. The analysis methods are also easy to follow with very clear equations. The authors present a very well-designed study with convincing evidence of the dynamic nature of heteroresistance, although with a specific mechanism and in a small sample set. The implications of this work are very important for antibiotic stewardship as well as for the detection and treatment of heteroresistant strains in a clinical setting.

I just have a couple of comments I think could be addressed:

- 1) In your discussion you correctly point out that your work emphasizes the need for improved detection methods – can you give some suggestions for what changes to existing methods or new assays could be developed that can better detect these minor resistant subpopulations? Similarly, what approaches could/should be taken for treating patients with heteroresistant infections to avoid enriching the resistant subpopulation?
- 2) You presumably have access to more heteroresistant isolates from your previous work (citation 8), is there a reason you focused on just *E. coli* and *S. Typhimurium*? Do these results hold up in other bacteria.

3) I found just 2 typographical errors

a. Line 79: "...how this dynamics affects..." should be "...how this dynamic affects..." or "...how these dynamics affect..." though I would prefer the former.

b. Figure 1, line 618-620: Inconsistencies in the placement of the lettered sub-headings. (A) is in boldface and at the beginning of the description, (B) is at both the beginning and the end of the description, (C) is at the end of the description, and (D) is in the middle of the description.

RESPONSE TO REVIEWER COMMENTS

Reviewer #1 (Remarks to the Author):

The authors present a timely and important manuscript focused on the dynamics of antibiotic resistant populations of cells exhibiting resistance gene amplifications. This group has been instrumental in showing that gene amplification leads to heteroresistance but the dynamics of these amplifications have not previously been studied systematically. This manuscript is viewed as being an important and foundational building block for the field.

We thank the reviewer for the positive comments.

Major comments

1) Figure 3A – there is a trend where the growth rate is lower with higher gene copy number. Are these curves statistically different?

This is an interesting point and the suggested statistical analysis was addressed. The results obtained can be found in Supplementary Table 2 and Supplementary Figure 4, and as can be seen, for concentrations below the MIC of the parental strain the resistant isolate with lower copy number grew faster than the isolate with higher copy number. When selective pressure for the resistance phenotype increased, we observed that the resistant isolate with higher copy number grew significantly faster than the isolate with lower copy number. We have included this analysis on the Results section (lines 140-145).

2) Is there a reason that the beta-lactams were not included in Figure 3?

Yes, this is because of the difficulties in determining the exponential growth rate in the presence of these antibiotics at sub-MIC levels. Growth dynamics where cell death is followed by bacterial “regrowth” in the presence of beta-lactam antibiotics have been widely observed in studies concerning growth in the presence of these drugs. This same dynamic was observed by us when studying the growth in the presence sub-lethal antibiotic concentrations of cefepime and piperacillin-tazobactam. Thus, we excluded the beta-lactams from fig. 3.

3) Are the MICs for each strain and the relevant antibiotic listed anywhere? It would be interesting to know how they relate to the breakpoint for that given antibiotic. It seems that these are all strains with MICs below the breakpoint.

We added the MICs for each strain in Supplementary Table 1. As stated, the MICs of these strains are all below the Clinical Breakpoint for the relevant antibiotic.

Minor comments

1) Is table 2 necessary? Every clone for each strain/antibiotic pair has the same amplification. The relevant information seems to be in Table 1 already.

We would like to keep this Table since it summarizes the information in a visual way.

2) The term “susceptible strain” is used often. This may be confusing or misleading to readers since the strains are known to contain resistant cells. Perhaps the term “parental strain” or “pre-treatment culture” would be better? It is important to choose a term that does not give the impression that there are no resistant cells present.

We apologize for the lack of a clear designation for the “susceptible strain” throughout the manuscript. From a formal point of view any culture with a high number of cells would contain some rare resistant cells and whether it is classified as susceptible or heteroresistant would then largely depend on the sensitivity of the diagnostic method. The isolates studied here are classified as susceptible by a standard method such as disk diffusion (simply because the method is not very sensitive for heteroresistance, see Fig 4) but a more sensitive method (like a PAP test) would classify them as HR. Anyhow, we agree with the reviewer that it is important to choose a term that does not mislead the readers and we have followed the suggestion to designate them as “parental strains”

3) Figure 2 – the y-axes say “resistant mutants”. This is not a critical point but are the authors referring to resistant cells? If so, are all the resistant cells considered mutants? If variation in copy number among individual cells is the standard for certain strains, maybe the resistant cells with higher copy number should be viewed as some form of variant rather than a mutant? In these cases there are no point mutations or acquisition of genes, but just a change in copy number of existing genes.

We would argue that also a copy number variant is a mutant since DNA content and sequence is changed.

Reviewer #2 (Remarks to the Author):

In this manuscript, the authors sought to examine the phenomenon of heteroresistance and its related dynamics. Understanding the heteroresistance dynamics, development and perpetuation of resistant subpopulations, will allow for better diagnostics and clinical treatment of patients harboring heteroresistant bacteria. The manuscript is addressing an important gap in the understanding of E.coli and S. enterica clinical disease and is well positioned to demonstrate dynamics under different drug pressures.

Overall this is a well executed study, addressing several aspects of heteroresistance dynamics in two different organisms and several different drugs illustrating possible far-reaching impacts. The

manuscript could be strengthened by streamlining sample processing, with so many different sections it is confusing what isolate (or evolution of the isolate) the experiments are run on. The addition of a sample workflow might help clarify, as there are several sections where partial sample is saved, and some extracted for analysis at several of the same time points. The introduction is well laid out and clear with 5 main objectives, the results section has 6 components. The manuscript could be strengthened by having the results broken up in the same 5 objectives described in the last paragraph of the introduction. Lastly, in the discussion section could be strengthened by referring back to tables and figures that contain the results that are being discussed. Overall, this manuscript utilizes several different laboratory approaches, resulting in a well supported description of the instability of heteroresistance in these two organisms.

We thank the reviewer for the insightful and constructive comments to the manuscript. A sample workflow with details about all the methods used for selection and analysis of the heteroresistant sub-populations is now included (Supplementary Figure 7) and referencing to this figure is included in the Materials section. The result section was rearranged in the five objectives described in the introduction to help the reader go through the main findings of this work. References to the tables and figures containing the results were added in the Discussion section.

Minor

1. Table 1 – add units on total DNA column heading

Corrected.

2. Table 3 – add definitions of R/S/- in table legend

Definitions of R/S/- were included in the table legend.

3. Figure 1 – add additional time points for the 1xMIC in figure legend

The information about the analysed time points was added in the figure legend.

4. Figure 3 – need table legend that describes the shorthand (CN= Copy number) and add color depictions for greater impact of figure

Completed.

5. Figure 4 – does not appear to be reference within manuscript text

The reference to Fig. 4 was added to the text.

6. Supplemental figures are not numbered in the order of appearance, sub fig 1 &2 are referenced in methods

Corrected.

7. Supplemental figure 4 – missing figure number and title

Corrected.

8. Supplemental figure 5 – add to figure legend what each color represents (yellow, blue, red, turquoise represent E. coli DA33135 resistant to tobramycin, S. Typhimurium DA34827 resistant to tetracycline, S. Typhimurium DA34833 resistant to cefepime and E. coli DA61218 resistant to piperacillin-tazobactam respectively.

The suggested information was added to the figure legend.

9. Result section - Within text can refer to Fig. 3 and remove AB designation (line 142 and 144)

This information was simplified by referring to Fig. 3.

10. Results section - Additionally add color into the description in the text to aid in reader's conclusion.

Something like – “ The crossover point of these two lines (gray versus colored lines in Fig. 3) is the MSC...” (line 143-144)

Completed.

Reviewer #3 (Remarks to the Author):

It is well known that heteroresistance – a mixed population of susceptible and resistant cells within a bacterial isolate – can cause treatment failure as the resistant subpopulation can outcompete the susceptible subpopulation in the presence of antibiotics and is a major public health concern with some bacterial pathogens. This study characterizes the dynamic nature of these subpopulations within four heteroresistant clinical isolates and provides convincing evidence that: 1) not only can the resistant subpopulations be enriched in the presence of antibiotic, but they can also be reduced in the absence of antibiotic pressure, and 2) the enrichment can occur with antibiotic levels far below the MIC. The study focuses on a specific mechanism of heteroresistance, “an increased copy number of resistance-conferring tandem gene amplifications that are intrinsically unstable and costly”. While my experience with heteroresistance is in *M. tuberculosis* which doesn't exhibit this mechanism, previous work by the authors has shown it to be the most common type of heteroresistance in gram-negative bacteria. The study tests multiple concentrations (down to 1/16x MIC) of antibiotic and time points (up to 160 generations) of bacterial growth, and get consistent results showing that even the lowest concentrations of antibiotic will increase the copy number of resistance genes within the population, and further show by two different methods that the increased copy number is associated with increased resistance. They then show that the copy number is reduced again using a serial passage experiment in the absence of antibiotic. Additionally, they characterized the structure and location of these tandem mutations and calculated the fitness cost of them, and also show that this instability in the population dynamics can confound the detection of resistance using the standard lab assay. While I am primarily a bioinformatician and can't speak to the specifics of the laboratory methods, they are very detailed and clearly worded which should make the results easy to reproduce. The analysis methods are also easy to follow with very clear equations. The authors present a very well-designed study with convincing evidence of the dynamic nature of heteroresistance, although with a specific mechanism and in a small sample set. The implications of this work are very important for antibiotic stewardship as well as for the detection and treatment of heteroresistant strains in a clinical setting.

I just have a couple of comments I think could be addressed:

1) In your discussion you correctly point out that your work emphasizes the need for improved detection methods – can you give some suggestions for what changes to existing methods or new assays could be

developed that can better detect these minor resistant subpopulations? Similarly, what approaches could/should be taken for treating patients with heteroresistant infections to avoid enriching the resistant subpopulation?

We thank the reviewer for raising these relevant questions. One important change to the existing methods used in clinics is considering maintaining antibiotic selection during the entire sampling period since we show that the instability of the heteroresistance phenotype and rapid loss of gene amplifications in the absence of selection can revert the resistance phenotype to susceptibility. As shown in the blood culture experiments, the lack of selection during antibiotic susceptibility testing leads to misdiagnosis of the resistance phenotype of heteroresistance isolates. We have added this information in the Discussion section (lines 303 - 304).

Regarding treatment, recent animal experiments from David Weiss lab at Emory (Band et al., Nat Microbiol 2019) have shown how combination therapy might allow eradication of the resistant sub-population. We have briefly commented on this in the Discussion (lines 305-308).

2) You presumably have access to more heteroresistant isolates from your previous work (citation 8), is there a reason you focused on just E. coli and S. Typhimurium? Do these results hold up in other bacteria.

It was mainly a practical reason to include a reasonable number of isolates and S. Typhimurium and E. coli are historically close to our hearts. In the previous work mentioned by the reviewer, it was shown for several clinical isolates of four different species that gene amplifications are the main cause for the heteroresistance phenotype. We would expect from these results that for other species, where the same conditions are present, similar dynamics will be seen. We are presently examining in K. pneumoniae and A. baumannii as well as several Gram-positives (S. aureus, S. pyogenes, S. pneumoniae and E. faecalis) to what extent these findings are generalizable.

3) I found just 2 typographical errors

a. Line 79: "...how this dynamics affects..." should be "...how this dynamic affects..." or "...how these dynamics affect..." though I would prefer the former.

Corrected.

b. Figure 1, line 618-620: Inconsistencies in the placement of the lettered sub-headings. (A) is in boldface and at the beginning of the description, (B) is at both the beginning and the end of the description, (C) is at the end of the description, and (D) is in the middle of the description.

Corrected.

REVIEWERS' COMMENTS:

Reviewer #1 (Remarks to the Author):

The authors have addressed all of my concerns.

There may be an error in the reference in the last sentence of the discussion.

Reviewer #2 (Remarks to the Author):

In this manuscript, the authors sought to examine the phenomenon of heteroresistance and its related dynamics. Understanding the heteroresistance dynamics, development and perpetuation of resistant subpopulations, will allow for better diagnostics and clinical treatment of patients harboring heteroresistant bacteria. The manuscript is addressing an important gap in the understanding of *E. coli* and *S. enterica* clinical disease and is well positioned to demonstrate dynamics under different drug pressures.

Overall, the response to reviewers has strengthened the manuscript describing a well executed study, addressing several aspects of heteroresistance dynamics in two different organisms and several different drugs illustrating possible farther reaching impacts.

Reviewer #3 (Remarks to the Author):

I am satisfied with the additions and corrections that have been made to this manuscript and feel it gives a well supported and clear picture of the dynamics of heteroresistance in *E. coli* and *S. Typhimurium* that was previously not well understood.